# Association of sleep quality and sleep duration with serum uric acid levels in adults

**Yu-Tsung Chou**[1,2], **Chung-Hao Li**[1,2], **Wei-Chen Shen**[1], **Yi-Ching Yang**[1,3], **Feng-Hwa Lu**[1,3], **Jin-Shang Wu**[1,3,4ʘ]*, **Chih-Jen Chang**[1,5ʘ]*

1 Department of Family Medicine, National Cheng Kung University Hospital, College of Medicine, National Cheng Kung University, Tainan, Taiwan, 2 Department of Health Management Center, National Cheng Kung University Hospital, College of Medicine, National Cheng Kung University, Tainan, Taiwan, 3 Department of Family Medicine, College of Medicine, National Cheng Kung University, Tainan, Taiwan, 4 Department of Family Medicine, National Cheng Kung University Hospital, Dou-Liou branch, College of Medicine, National Cheng Kung University, Yunlin, Taiwan, 5 Department of Family Medicine, Ditmanson Medical Foundation Chia-Yi Christian Hospital, Chiayi, Taiwan

ʘ These authors contributed equally to this work.
* jins@mail.ncku.edu.tw (JSW); changcj.ncku@gmail.com (CJC)

**Data Availability Statement:** All relevant data are within the Supporting Information files.

**Funding:** This research was funded by the National Cheng Kung University Hospital, Taiwan (NCKUH-10903022). The funders had no role in study

## Abstract

### Objective

To date, the association between sleep duration or sleep quality and hyperuricemia has remained unclear. In addition, sleep duration and quality were not considered concomitantly in previous studies. Thus, this study was aimed toward an examination of the association of sleep duration and quality with uric acid level in a Taiwanese population.

### Methods

A total of 4,555 patients aged ≥18 years were enrolled in this study. The sleep duration was classified into three groups: short (<7 h), normal (7–9 h), and long (≥9 h). The Pittsburgh Sleep Quality Index (PSQI) was used to evaluate sleep quality, and poor sleep quality was defined as a global PSQI score of >5.

### Results

Poor sleepers were younger and had lower body mass index, blood pressure, uric acid, blood sugar, cholesterol, creatinine level, shorter sleep duration, and engaged in less exercise but had a higher white blood cell count and prevalence of smoking as compared to good sleepers. There were also differences in body mass index, blood pressure, uric acid, blood sugar, lipid profiles, and sleep quality among subjects with different sleep durations. After adjusting for other variables, poor sleep quality was associated with lower uric acid levels. In addition, short sleep duration was positively associated with higher uric acid levels.

### Conclusions

Poor sleep quality was related to lower uric acid levels, whereas short sleep duration was associated with higher uric acid levels.

design, data collection and analysis, decision to publish, or preparation of the manuscript.

**Competing interests:** The authors have declared that no competing interests exist.

## Introduction

Hyperuricemia is one of most common metabolic disorders in modern society. Its prevalence is approximately 20% worldwide [1], and it may result in several crystal deposition-related disorders, such as gout, urate nephropathy, and urolithiasis [2]. Furthermore, hyperuricemia is also related to specific medical conditions, such as chronic kidney disease [3, 4], cardiovascular disease [5, 6], hypertension [7, 8], and early mortality [9]. In addition, some studies have demonstrated that high-normal levels of serum uric acid are also associated with risk of diabetes [10], impaired renal function [11] and even mild cognitive impairment [12]. However, low uric acid levels have been shown to be related to a higher prevalence of Alzheimer's disease, Parkinson's disease (PD), and PD-related diseases such as multiple system atrophy and progressive supra-nuclear palsy [13–16]. Some studies have shown the beneficial aspects of uric acid for its antioxidative and neuroprotective effects [17–21], and the optimal uric acid level appears to be beneficial for health [19].

Sleep plays an important role in the maintenance of personal health. However, poor sleep quality and sleep deprivation are common nowadays [22]. Various studies have demonstrated that inadequate sleep, such as short sleep duration or poor sleep quality, are associated with cardiovascular disease, obesity, hypertension, non-alcoholic liver disease, diabetes mellitus (DM), dyslipidemia, and even all-cause mortality [23–31].

Physiologically, sleep mediates the level of catecholamine and cortisol [32, 33], which may potentially influence uric acid levels [34]. Most previous studies have found that sleep-disordered breathing such as obstructive sleep apnea (OSA) is associated with hyperuricemia and gout [35–38]. However, studies investigating the relationship between sleep quality or sleep duration and uric acid levels are limited [39–44]. One study found lower serum uric acid levels to be associated with poorer sleep quality [39]. Another study showed that participants had a positive association between sleep quality and uric acid level after acute ischemic stroke [40]. As for the relationship between sleep duration and uric acid, two studies demonstrated that sleep duration was inversely associated with serum uric acids [41, 42], and another study found that long sleepers had a lower prevalence of hyperuricemia [43]. However, the relationship between sleep duration and uric acid was insignificant in another study based on the National Health and Nutrition Examination Survey (NHANES) database [44]. To our knowledge, there have been no studies considering the concomitant influence of both sleep quality and sleep duration on uric acid levels. Thus, this study was aimed toward an evaluation of the association of sleep duration and sleep quality with uric acid levels in an adult population.

## Methods

### Study population

A total of 4,592 patients who visited a health center for a self-motivated general health assessment were recruited in this study. Data were obtained from a health examination center at National Cheng Kung University Hospital, from October 2001 to August 2009. Secondary data without any personal identifiable information was used, and the study was approved by the Institutional Review Board of NCKUH in Taiwan (IRB number: A-ER-107-285). Participants aged <18 years, on medications for hyperuricemia, hypertension, DM or dyslipidemia, and those with incomplete data were excluded.

### Clinical measurements

The baseline data included demographic information, personal medical and medication history, lifestyle habits (alcohol consumption, cigarette smoking, and regular exercise), and sleep

status. Current alcohol consumption was defined as at least one alcoholic drink per week in the past 6 months. Current smoking was defined as at least 20 cigarettes per month in the past 6 months. Regular exercise was defined as a habit of engaging in vigorous exercise at least 3 times per week. Sleep duration and quality were assessed using the Chinese version of the PSQI [45], which was used to evaluate the participants' sleep quality over a 1-month time interval using a self-rated questionnaire. The Chinese version of the PSQI was validated as an assessment tool for evaluating sleep condition, with an overall reliability coefficient of 0.82–0.83, and good test-retest reliability, with a coefficient of 0.85. A Chinese version of the PSQI global score > 5 had a diagnostic sensitivity of 98% and specificity of 55% in terms of distinguishing poor and good sleepers [45]. The PSQI reflects seven important aspect of sleep, including sleep duration, habitual sleep efficiency, sleep latency, subjective sleep quality, use of sleep medications, sleep disturbances, and presence of daytime dysfunction. The total global PSQI scores range from 0 to 21 points, where a lower global PSQI score indicates better sleep quality. Participants with a global PSQI score of >5 were defined as poor sleepers. Sleep duration was categorized into three groups: short (<7 h/day), normal (7–9 h/day), and long (≧9 h/day) [46].

Each participant's body weight and height were measured in light indoor clothing. The body mass index (BMI) was calculated as weight (kg)/height squared ($m^2$). Obesity was defined as BMI $\geq$ 27kg/$m^2$ according to the guidelines suggested by the Department of Health in Taiwan [47]. After resting for at least 5 min, the blood pressure of the right branchial artery of each participant was measured in the supine position. Hypertension was defined as a right brachial systolic blood pressure (SBP) of $\geq$140 mmHg, a diastolic blood pressure (DBP) $\geq$ 90 mmHg [48], or a positive history of hypertension.

After fasting for at least 12 h, all participants underwent blood sampling for basic biochemical examinations. All of the blood samples were drawn from 7 am to 9 am in the morning. None of the women were pregnant when tested. Laboratory data including serum uric acid, FPG, HbA1C, 2-hPG, total cholesterol (TC), triglyceride (TG), high-density lipoprotein cholesterol (HDL-C), alanine aminotransferase (ALT), white blood cell (WBC) count, and creatinine levels were collected. For all participants without a medical history of diabetes, a 75-g oral glucose tolerance test was performed after overnight fasting for 12 h, with a normal diet for 3 days before the test and abstaining from smoking for more than one day. DM was defined as a self-reported DM history, 2-h postprandial glucose (2-hPG) of ≧200 mg/dL, fasting plasma glucose (FPG) of ≧126 mg/dL, or hemoglobin A1c (HbA1c) of ≧6.5% according to the American Diabetes Association's diagnostic criteria [49]. The estimated glomerular filtration rate (eGFR) was calculated using the Chronic Kidney Disease Epidemiology Collaboration (CKD-EPI) equation [50].

## Statistical analysis

The data analysis was performed using SPSS software (version 17.0, SPSS, Inc., Chicago, IL). In the univariate analysis, an independent t-test was performed for the continuous variables and a Pearson's chi-square analysis was used for the categorical variables to compare the participants' data based on sleep quality. Sleep durations was categorized into three groups, and a Pearson's chi-square analysis and an analysis of variance were used, where appropriate. In the multivariate analysis, linear regressions were performed to investigate the association of sleep status, including sleep quality and duration, with uric acid levels. Adjusted variables included age, gender, obesity, estimated glomerular filtration rate (eGFR) <60, TC/HDL-C ratio, WBC count, DM, HTN, current smoking, alcohol consumption, and regular exercise. Statistical significance was defined as $p < 0.05$.

## Results

Of all 4,592 participants recruited for the final analysis, 59.9% were poor sleepers. Table 1 shows a comparison of the participants' clinical characteristics based on sleep quality. There were significant differences between the good and poor sleepers in terms of age, gender, BMI, blood pressure, uric acid, FPG, TC, WBC count, and renal functions. Poor sleepers had shorter sleep duration, engaged in less exercise, and had a higher prevalence of current smoking. Table 2 shows a comparison of the participants' clinical characteristics based on sleep duration. There were differences in body mass index, diastolic blood pressure, renal function, uric acid, lipid profiles, and sleep quality scores. The prevalence of poor sleepers, smoking, and exercise habits were also different among subjects with different sleep duration.

During the linear regression analysis, we initially investigated the relationships between uric acid levels and sleep quality and duration, separately. The results revealed that poor sleep quality was associated with lower uric acid levels, whereas short sleep duration was related to higher uric acid levels. Subsequently, we analyzed the relationships between both sleep quality and duration and uric acid levels using a multiple linear regression model. As shown in Table 3, when adjusted for sleep duration, poor sleepers remained associated with lower uric

**Table 1. Comparisons of participants' clinical characteristics based on sleep quality.**

| Variables | PSQI ≤ 5 (n = 1842) | PSQI > 5 (n = 2750) | P value |
|---|---|---|---|
| Age, years | 44.8 ± 11.1 | 43.4 ± 11.8 | <0.001 |
| Male | 1156 (62.8) | 1619 (58.9) | 0.008 |
| BMI, kg/m$^2$ | 24.0 ± 3.3 | 23.9 ± 3.6 | 0.202 |
| BMI ≥ 27 | 299 (16.2) | 465 (16.9) | 0.546 |
| SBP, mmHg | 114.7 ± 15.1 | 113.8 ± 15.2 | 0.034 |
| DBP, mmHg | 68.1 ± 10.1 | 67.4 ± 10.4 | 0.038 |
| FPG, mg/dL | 90.7 ± 18.6 | 89.6 ± 19.8 | 0.048 |
| ALT, U/L | 31.7 ± 31.0 | 32.0 ± 34.2 | 0.765 |
| Cholesterol, mg/dL | 195.6 ± 35.3 | 192.6 ± 36.5 | 0.006 |
| Triglyceride, mg/dL | 122.8 ± 85.9 | 123.7 ± 83.8 | 0.724 |
| HDL-C, mg/dL | 49.8 ± 13.3 | 49.4 ± 13.4 | 0.355 |
| Cholesterol/HDL-C | 4.2 ± 1.3 | 4.2 ± 1.3 | 0.584 |
| Creatinine, mg/dL | 0.88 ± 0.18 | 0.86 ± 0.18 | 0.002 |
| eGFR <60 | 170 (9.2) | 203 (7.4) | 0.025 |
| WBC count, 10^3/μL | 6.1 ± 1.6 | 6.2 ± 3.0 | 0.013 |
| Uric acid, mg/dL | 6.1 ± 1.5 | 6.0 ± 1.5 | 0.038 |
| Hypertension | 150 (8.1) | 247 (9.0) | 0.322 |
| Diabetes mellitus | 109 (5.9) | 172 (6.3) | 0.640 |
| PSQI score | 3.9 ± 1.1 | 8.7 ± 2.6 | <0.001 |
| Sleep duration, h/day | 7.1 ± 0.8 | 6.1 ± 1.1 | <0.001 |
| <7 | 527 (28.6) | 1959 (71.2) | <0.001 |
| 7–9 | 1266 (70.8) | 727 (28.5) | |
| >9 | 49 (2.7) | 24 (0.9) | |
| Current alcohol use | 284 (15.4) | 477 (17.3) | 0.085 |
| Current smoking | 267 (14.5) | 494 (18.0) | 0.002 |
| Exercise ≥ 3/wk | 265 (14.4) | 295 (10.7) | <0.001 |

Data expressed as mean ± standard deviation or number (percent).

SBP: systolic blood pressure, DBP: diastolic blood pressure, FPG: fasting plasma glucose, ALT: Alanine Aminotransferase, HDL-C, high-density lipoprotein-cholesterol, eGFR: estimated glomerular filtration rate, PSQI: Pittsburgh Sleep Quality Index

**Table 2. Comparisons of participants' clinical characteristics based on sleep duration.**

| Variables | Sleep duration < 7 h/day (n = 2486) | Sleep duration 7–9 h/day (n = 2087) | Sleep duration ≥ 9 h/day (n = 19) | P value |
|---|---|---|---|---|
| Age, years | 45.0 ± 11.6 | 43.0 ± 11.2 | 38.1 ± 13.9 | <0.001 |
| Male | 1490 (59.9) | 1249 (61.4) | 36 (49.3) | 0.087 |
| BMI, kg/m$^2$ | 24.1 ± 3.5 | 23.8 ± 3.4 | 23.1 ± 3.9 | 0.003 |
| BMI ≥ 27 | 449 (18.1) | 405 (15.0) | 10 (13.7) | 0.018 |
| SBP, mmHg | 114.4 ± 15.3 | 114.0 ± 14.9 | 111.5 ± 15.3 | 0.225 |
| DBP, mmHg | 67.9 ± 10.5 | 67.5 ± 10.0 | 64.1 ± 10.5 | 0.003 |
| FPG, mg/dL | 90.0 ± 20.0 | 90.0 ± 18.0 | 92.6 ± 30.5 | 0.527 |
| ALT, U/L | 32.0 ± 31.5 | 31.8 ± 35.0 | 28.6 ± 23.3 | 0.691 |
| Cholesterol, mg/dL | 194.9 ± 37.2 | 192.5 ± 34.8 | 193.4 ± 34.1 | 0.078 |
| Triglyceride, mg/dL | 124.8 ± 84.4 | 121.3 ± 83.9 | 129.3 ± 106.6 | 0.322 |
| HDL-C, mg/dL | 49.6 ± 13.5 | 49.4 ± 13.1 | 51.6 ± 14.4 | 0.370 |
| Cholesterol/HDL-C | 4.2 ± 1.3 | 4.2 ± 1.3 | 4.1 ± 1.4 | 0.421 |
| Creatinine, mg/dL | 0.87 ± 0.18 | 0.87 ± 0.18 | 0.86 ± 0.18 | 0.760 |
| eGFR <60 | 205 (8.2) | 165 (8.1) | 3 (4.1) | 0.443 |
| WBC count, 10^3/μL | 6.1 ± 1.6 | 6.2 ± 3.4 | 6.4 ± 1.8 | 0.301 |
| Uric acid, mg/dL | 6.1 ± 1.5 | 6.0 ± 1.5 | 6.0 ± 1.6 | 0.462 |
| Hypertension | 231 (9.3) | 160 (7.9) | 6 (8.2) | 0.237 |
| Diabetes mellitus | 163 (6.6) | 112 (5.5) | 6 (8.2) | 0.258 |
| Total PSQI | 8.1 ± 3.2 | 5.1 ± 2.3 | 4.9 ± 2.6 | <0.001 |
| PSQI >5 | 1959 (78.8) | 767 (37.7) | 24 (32.9) | <0.001 |
| Sleep duration, h/day | 5.7 ± 0.8 | 7.4 ± 0.5 | 9.3 ± 0.6 | <0.001 |
| Current alcohol use | 405 (16.3) | 344 (16.9) | 12 (16.4) | 0.851 |
| Current smoking | 409 (16.5) | 338 (16.6) | 14 (19.2) | 0.823 |
| Exercise ≥ 3/wk | 310 (12.5) | 243 (12.0) | 7 (9.6) | 0.687 |

Data expressed as mean ± standard deviation or number (percent).

BMI: body mass index, SBP: systolic blood pressure, DBP: diastolic blood pressure, FPG: fasting plasma glucose, ALT: Alanine Aminotransferase, TC: total cholesterol, HDL-C, high-density lipoprotein-cholesterol, eGFR: estimated glomerular filtration rate, PSQI: Pittsburgh Sleep Quality Index

acid levels [b-coefficient: -0.085, 95% confidence interval (CI): −0.161 to −0.008, $p$ = 0.030], and short sleep duration was associated with higher serum uric acid levels (b-coefficient: 0.106, 95% CI: 0.031–0.181, $p$ = 0.006). However, no association was found between long sleeper duration and uric acid levels. In addition, uric acid levels were positively associated with overweight and obesity, the TC/HDL-C ratio, WBC count, hypertension, and alcohol consumption habits, but decreased eGFR levels. In addition, age, DM, and smoking were negatively associated with uric acid levels. In the linear regression model, we observed no multicollinearity among the covariates, with a variance inflation factor < 1.4.

## Discussion

In this study, poor sleep quality was related to lower uric acid levels, and short sleep duration was associated with a higher uric acid levels after adjusting for confounders. To the best of our knowledge, this is the first study that concomitantly investigated the association of both sleep quality and sleep duration with uric acid levels. There have been several studies discussing sleep quality and uric acid levels, but sleep duration was not taken into consideration [39, 40, 44, 51]. One case-control study showed that uric acid levels were negatively related to PSQI scores, indicating reduced uric acid levels in subjects with poor sleep quality [39]. Another study conducted with ischemic stroke patients also showed reduced uric acid levels in

**Table 3. Associations between sleep quality, sleep duration, and uric acid levels based on the linear regression model.**

| Variables | Univariate | | | Multivariate | | |
|---|---|---|---|---|---|---|
| | β | 95% CI | p-value | β | 95% CI | p-value |
| PSQI scale | | | | | | |
| ≥ 5 vs < 5 | −0.096 | −0.187 ~ −0.005 | 0.038 | −0.085 | −0.161 ~ −0.008 | 0.030 |
| sleep duration | | | | | | |
| < 7 h vs 7–9 h | 0.055 | −0.035 ~ 0.145 | 0.231 | 0.106 | 0.031 ~ 0.181 | 0.006 |
| ≥ 9 h vs 7–9 h | −0.029 | −0.388 ~ 0.330 | 0.874 | 0.164 | −0.107 ~ 0.436 | 0.236 |
| Age, years | | | | −0.009 | −0.012 ~ −0.006 | <0.001 |
| Sex, male vs female | | | | 1.401 | 1.321 ~ 1.480 | <0.001 |
| Obesity, yes vs no | | | | 0.518 | 0.423 ~ 0.614 | <0.001 |
| Hypertension, yes vs no | | | | 0.227 | 0.103 ~ 0.351 | <0.001 |
| Diabetes mellitus, yes vs no | | | | −0.270 | −0.042 ~ −0.125 | <0.001 |
| TC/HDL ratio | | | | 0.235 | 0.205 ~ 0.264 | <0.001 |
| eGFR <60 yes vs no | | | | −0.889 | −1.021 ~ −0.756 | <0.001 |
| WBC count, 10^3/μL | | | | 0.035 | 0.022 ~ 0.049 | <0.001 |
| Current alcohol use, yes vs no | | | | 0.216 | 0.120 ~ 0.313 | <0.001 |
| Current smoking, yes vs no | | | | −0.141 | −0.241 ~ −0.042 | 0.005 |
| Exercise habit, yes vs no | | | | −0.005 | −0.099 ~ 0.109 | 0.922 |

Interaction between sleep duration and sex, $p = 0.157$

Interaction between sleep quality and sex, $p = 0.364$

Interaction between sleep duration and sleep quality, $p = 0.153$

CI: confidence interval, PSQI: Pittsburgh Sleep Quality Index, BMI: body mass index, TC: total cholesterol, HDL-C, high-density lipoprotein-cholesterol, eGFR: estimated glomerular filtration rate

individuals with poor sleep quality [40]. However, one study, based on the National Health and Nutrition Examination Survey (NHANES) showed that sleep quality, as analyzed by the Sleep Disorders Questionnaire, was not associated with uric acid levels [51]. The discrepancies in the observed relationships between sleep quality and uric acid levels described in these studies may be related to the differences in population ethnicities, definition of sleep quality, and other potentially adjusting confounders, such as medications for chronic diseases, including diabetes, hypertension, dyslipidemia, and hyperuricemia. In the present study, the population was free from the influence of important cardiometabolic medications, and we concurrently considered the influences of both sleep quality and duration, while carefully adjusting for important confounders. As a result, we found that poor sleep quality was associated with lower uric acid levels.

The possible mechanism for the relationship between sleep quality and uric acid remains unclear. Since sleep disturbances are related to increased systemic inflammation and oxidative stress [32], the beneficial aspects of uric acid may play a role among good sleepers. Uric acids have been to have antioxidative and neuroprotective effects [14, 17, 19, 21, 52]. A study conducted by Bowman et. al showed that uric acid is an important endogenous antioxidant in the central nervous system [53]. One Korean study revealed that higher uric acids were related to better antioxidative capacity [54]. Previous studies also elucidated that poor sleep is related to increased oxidative stress in both animal and human models [55, 56]. In addition, higher oxidative stress accompanied by poor sleep may cause a greater antioxidative reaction [34] between uric acid and reactive oxygen species, such as reactions with peroxynitrite or chelation of metal ions [17, 57]. These reactions may cause consumption of uric acid as an anti-oxidant, which may in turn result in lower uric acid levels among poor sleepers [17].

In terms of sleep duration, the results of this study showed that short duration sleepers had higher uric acid levels. Our results were similar to other studies demonstrating an association between short sleep duration and hyperuricemia [41, 42, 44]. The association between short sleep duration and hyperuricemia may be caused by cardiometabolic disorders since studies have shown that short sleep duration increases the risks associated of obesity, metabolic syndrome, and hypertension [58–60], which are all common risk factors for hyperuricemia. Although we adjusted for obesity, TC/HDL-C ratio, WBC count, DM, and HTN, short sleep duration remained associated with hyperuricemia. Thus, further potential mechanisms for this relationship must be clarified. The literature has shown that sleep loss is associated with elevated catecholamine levels [61], which may result in increased nucleotide turnover and enhanced production of endogenous uric acid [34]. Furthermore, uric acids act as antioxidants, especially in a hydrophilic environment [17]. However, obesity and metabolic syndrome have shown to be associated with a more hydrophobic environment, which is rich in lipids, resulting in an unfavorable status for the antioxidative effects of uric acid [17, 52]. Thus, uric acid may be relatively limited in its role as an antioxidant and thus may be less likely to be metabolized, resulting in elevated uric acid levels among subjects with short sleep durations.

Our study also found an insignificant relationship between long sleep duration and uric acid levels, when considering sleep quality and other confounding factors. A study based on data from NHANES, from 2005–2006 and 2007–2008, showed no association between long sleep duration and hyperuricemia, whereas a study performed examining an elderly Mediterranean population at high cardiovascular risk revealed that long sleepers had lower uric acid levels [41, 44]. Therefore, the association between long sleep duration and uric acid level remains inconclusive. The inconsistent results between this study and two other studies [33, 36] may be related to subject selection, such as differences in age, race, inclusion criteria, and covariate adjustments. In addition, the sample size of patients with long sleep duration was relatively small in this study (n = 19) and in another study examining an elderly Mediterranean population (n = 37) [36]. Further studies with larger sample sizes are necessary to clarify the relationship between long sleep duration and uric acid levels.

In this study, the serum uric acid level was also positively related to male gender, BMI, hypertension, lipid profile, decreased eGFR, WBC count and alcohol consumption. These results were consistent with those of previous studies [62, 63]. It is well known that males have higher uric acid level than females [64]. Obesity, dyslipidemia, hypertension, and renal insufficiency result in decreased renal excretion of urate and thus cause increased serum uric acid levels [63]. In addition, obesity, and dyslipidemia also affect uric acid levels through elevated insulin resistance and increased urate production [65, 66]. Also, obesity is related to increased visceral fat accumulation, which may result in increased plasma free fatty acids in hepatic tissue and subsequently lead to elevated uric acid production [67]. Higher WBC counts provoke production of tumor necrosis factor-alpha and interleukin-6, which may lead to impaired insulin sensitivity and in turn to higher uric acid levels [68, 69]. As for the negative association between current smoking and uric acid levels, our findings were compatible with those of previous studies. The possible mechanism for this may be (1) inhibition of xanthine oxidase by cyanide during cigarette smoking and (2) increased oxidative stress during tobacco consumption, which results in the depletion of uric acid due to its antioxidative effects [70, 71]. A negative association between uric acid level and diabetes was also observed in this study, which was similar to the findings of a study conducted by Bandaru et al. This inverse relationship may be related to decreased uric acid resorption in the proximal tubule under hyperglycemic status in participants with diabetes [72]. Our study also revealed that uric acid level is inversely related to age. Another study from Chiou et al. showed the same result in a Taiwanese population [73], which found the youngest male group had the highest uric acid level. The opposite

association between age and uric acid may be related to lifestyle change, such as dietary and exercise habits, as well as a rapid increase in the prevalence of obesity and overweight among the younger generation in Taiwan [74].

Despite the strength of considering sleep duration, sleep quality, and other important covariates of hyperuricemia concomitantly, some limitations still exist in the present study. First, clarifying the causal relationship between sleep status and uric acid levels is difficult under a cross-sectional design. Second, participants with OSA, which is associated with hyperuricemia and gout [35, 37], was excluded only by self-reported history, due to the unavailability of subjective measurements such as polysomnography, and therefore, its influence may have been underestimated. Third, our participants' sleep quality and duration were obtained from self-reported questionnaires but not measured using objective medical devices, such as polysomnography or sleep actigraphy. Fourth, the sample size of long sleepers was relatively small, and the relationship between long sleep duration and uric acid levels was underestimated. Fifth, the effects of sleep quality on uric acid were relatively small, but significant, in the present study. These results provide a research direction for further studies exploring the association between sleep and uric acid to clarify their relationship. Finally, socioeconomic status [75] and dietary habits were not examined in this study.

In conclusion, poor sleep quality was found to be related to lower uric acid levels, whereas short sleep duration was associated with higher uric acid levels. The relationship between long sleep duration and uric acid was, however, insignificant.

## Supporting information

**S1 Data.**
(XLS)

## Acknowledgments

We are grateful to Prof. Chung-Yi Li for providing statistical consulting services from the Biostatistics Consulting Center, Clinical Medicine Research Center, National Cheng Kung University Hospital. The authors thank Enago (https://www.enago.tw/) for assistance with editing the manuscript. We also thank Savana Moore (Foreign Language Center, College of Liberal Arts, National Cheng Kung University) for the assistance of proofreading the revised manuscript.

## Author Contributions

**Conceptualization:** Yu-Tsung Chou, Wei-Chen Shen.

**Data curation:** Yu-Tsung Chou, Chung-Hao Li.

**Formal analysis:** Yu-Tsung Chou.

**Investigation:** Yu-Tsung Chou, Yi-Ching Yang, Feng-Hwa Lu.

**Methodology:** Yu-Tsung Chou, Chung-Hao Li, Wei-Chen Shen, Yi-Ching Yang.

**Project administration:** Yu-Tsung Chou, Jin-Shang Wu, Chih-Jen Chang.

**Resources:** Yu-Tsung Chou, Yi-Ching Yang, Feng-Hwa Lu, Jin-Shang Wu, Chih-Jen Chang.

**Software:** Yu-Tsung Chou.

**Supervision:** Jin-Shang Wu, Chih-Jen Chang.

**Validation:** Jin-Shang Wu, Chih-Jen Chang.

**Visualization:** Chih-Jen Chang.

**Writing – original draft:** Yu-Tsung Chou.

**Writing – review & editing:** Jin-Shang Wu, Chih-Jen Chang.

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
