## [Decision Letter · Decision Letter 0]

29 Jun 2020

PONE-D-20-14422

Association of sleep quality and sleep duration with serum uric acid level in adults

PLOS ONE

Dear Dr Chang

Thank you for submitting your manuscript to PLOS ONE. After careful consideration, we feel that it has merit but does not fully meet PLOS ONE’s publication criteria as it currently stands. Therefore, we invite you to submit a revised version of the manuscript that addresses the points raised during the review process.

We look forward to receiving your revised manuscript.

Kind regards,

Xianwu Cheng, M.D., Ph.D., FAHA

Academic Editor

PLOS ONE

Journal Requirements:

Additional Editor Comments (if provided):

Although the topic is interesting, as you will gather from the reviews, the referees identified substantive methodological problems, statistical analysis, and data presentation as well as the recruitment of all subjects. The editorial broad member also concurs. You may resubmit a revised version but it will be re-reviewed and there exists no guarantee that even with revision it will necessarily be accepted.

Reviewers' comments:

Reviewer's Responses to Questions

**Comments to the Author**

1. Is the manuscript technically sound, and do the data support the conclusions?

Reviewer #1: Yes

Reviewer #2: Yes

2. Has the statistical analysis been performed appropriately and rigorously? 

Reviewer #1: Yes

Reviewer #2: Yes

3. Have the authors made all data underlying the findings in their manuscript fully available?

Reviewer #1: Yes

Reviewer #2: Yes

4. Is the manuscript presented in an intelligible fashion and written in standard English?

Reviewer #1: Yes

Reviewer #2: Yes

5. Review Comments to the Author

Reviewer #1: In this manuscript, the authors investigate the relationship between uric acid levels and both sleep duration and sleep quality, using data from the PSQI. They found that decreases in sleep duration are associated with a higher UA level while poor sleep quality was associated with a lower UA.

While the paper is well-written and the statistics appear appropriate, I am unsure of the clinical significance of the statistical finding. In particular, if you look at Table 1, the UA difference is only 0.1 mg/dL between the two groups. This is statistically difference but not sure it is clinically significant, since both levels are likely in normal range. Then, if you look at Table 3, it would appear that the differences in UA between the two groups, after various corrections, remains quite small. Thus, it is not clear to me that the statistical difference in UA has any clinical outcomes that are important.

Therefore, the paper would be greatly enhanced if the authors had any outcome data related to UA that showed that the small differences make a clinical difference.

Reviewer #2: This article is a report of regression on a large dataset exploring the replationship between uric acid levels and duration of sleep. This article will contribute meaningful evidence to the scientific literature. It appears to be scieintifically developed and a good discussion of the reasons for the relatinship are discussed.

I am recommending that the authors consider adding one variable to their dataset that could be a mediator to the relationship. The authors will also need to edit the grammar and sentence structure in below mentioned sections of the paper.

Introduction:

Line 45: typo “lover”

Line 47: poor grammar/sentence structure

There is no mention of the possible physiological relationship between sleep duration and uric acid levels. This must be delineated to substantiate the scientific underpinnings of the study.

Study Population:

Poor grammar and sentence structure.

Why did you exclude medications? Again, poor writing.

Clinical measurements:

There is no mention of the validity and reliability of the PSQI sleep duration question. What construct is the sleep duration question really accessing? Is subjective report of sleep a valid measure? This must be substantiated.

Validation and measurement specifications and moderators of the uric acid test are not discussed. For example, does diet the day of the test have an impact on the uric acid level? How about time of day the blood was drawn? Was there only one data point for the uric acid level?

Results and Analysis:

Ok, although the PSQI is a valid screen for obstructive sleep apnea. Why didn’t you look at the relationship between OSA and uric acid levels? Do you have any measures of inflammation in your dataset? If so, you might want to look at that relationship.

Discussion:

Really nice review of the literature and discussion, but I think you are missing the point about OSA/obesity/uric acid levels.

6. PLOS authors have the option to publish the peer review history of their article (what does this mean?). If published, this will include your full peer review and any attached files.

Reviewer #1: **Yes: **James A Rowley, MD

Reviewer #2: **Yes: **Carla R. Jungquist

---

## [Author Response · Author response to Decision Letter 0]

14 Aug 2020

Thank you very much for your kind letter, along with the constructive comments of the reviewers concerning our manuscript. We have thoroughly considered all the comments of the reviewers and substantially revised our manuscript, and the major revised portions are marked in our revised manuscript. We also respond point by point to the reviewer’s comments as listed in the “Response to Reviewers”, along with a clear indication of the location of the revision. We look forward to hearing from you.

---

## [Decision Letter · Decision Letter 1]

2 Sep 2020

Association of Sleep Quality and Sleep Duration with Serum Uric Acid Levels in Adults

PONE-D-20-14422R1

Dear Dr Chang

We’re pleased to inform you that your manuscript has been judged scientifically suitable for publication and will be formally accepted for publication once it meets all outstanding technical requirements.

Kind regards,

Xianwu Cheng, M.D., Ph.D., FAHA

Academic Editor

PLOS ONE

Additional Editor Comments (optional):

Although the original reviewer#2 has delined to review second peer review process (minor revision), all original concerns have been addressed by the authors.

Reviewers' comments:

Reviewer's Responses to Questions

**Comments to the Author**

1. If the authors have adequately addressed your comments raised in a previous round of review and you feel that this manuscript is now acceptable for publication, you may indicate that here to bypass the “Comments to the Author” section, enter your conflict of interest statement in the “Confidential to Editor” section, and submit your "Accept" recommendation.

Reviewer #1: All comments have been addressed

2. Is the manuscript technically sound, and do the data support the conclusions?

Reviewer #1: Yes

3. Has the statistical analysis been performed appropriately and rigorously? 

Reviewer #1: Yes

4. Have the authors made all data underlying the findings in their manuscript fully available?

Reviewer #1: Yes

5. Is the manuscript presented in an intelligible fashion and written in standard English?

Reviewer #1: Yes

6. Review Comments to the Author

Reviewer #1: (No Response)

7. PLOS authors have the option to publish the peer review history of their article (what does this mean?). If published, this will include your full peer review and any attached files.

Reviewer #1: No

---

## [Editor Report · Acceptance letter]

8 Sep 2020

PONE-D-20-14422R1 

Association of Sleep Quality and Sleep Duration with Serum Uric Acid Levels in Adults 

Dear Dr. Chang:

I'm pleased to inform you that your manuscript has been deemed suitable for publication in PLOS ONE. Congratulations! Your manuscript is now with our production department. 

Kind regards, 

on behalf of

Associate Prof. Xianwu Cheng 

Academic Editor

PLOS ONE